# Using cross-species co-expression to predict metabolic interactions in microbiomes

Robert A. Koetsier,[1] Zachary L. Reitz,[1] Clara Belzer,[2] Marc G. Chevrette,[3] Jo Handelsman,[3] Yijun Zhu,[4] Justin J. J. van der Hooft,[1,5] Marnix H. Medema[1]

**ABSTRACT** In microbial ecosystems, metabolic interactions are key determinants of species' relative abundance and activity. Given the immense number of possible interactions in microbial communities, their experimental characterization is best guided by testable hypotheses generated through computational predictions. However, widely adopted software tools—such as those utilizing microbial co-occurrence—typically fail to highlight the pathways underlying these interactions. Bridging this gap will require methods that utilize microbial activity data to infer putative target pathways for experimental validation. In this study, we explored a novel approach by applying cross-species co-expression to predict interactions from microbial co-culture RNA-sequencing data. Specifically, we investigated the extent to which co-expression between genes and pathways of different bacterial species can predict competition, cross-feeding, and specialized metabolic interactions. Our analysis of the Mucin and Diet-based Minimal Microbiome (MDb-MM) data yielded results consistent with previous findings and demonstrated the method's potential to identify pathways that are subject to resource competition. Our analysis of the Hitchhikers of the Rhizosphere (THOR) data showed links between related specialized functions, for instance, between antibiotic and multidrug efflux system expression. Additionally, siderophore co-expression and further evidence suggested that increased siderophore production of the *Pseudomonas koreensis* koreenceine BGC deletion-mutant drives siderophore production in the other community members. In summary, our findings confirm the feasibility of using cross-species co-expression to predict pathways potentially involved in microbe-microbe interactions. We anticipate that the approach will also facilitate the discovery of novel gene functions through their association with other species' metabolic pathways, for example, those involved in antibiotic response.

**IMPORTANCE** An improved mechanistic understanding of microbial interactions can guide targeted interventions or inform the rational design of microbial communities to optimize them for applications such as pathogen control, food fermentation, and various biochemical processes. Existing methodologies for inferring the mechanisms behind microbial interactions often rely on complex model-building and are, therefore, sensitive to the introduction of biases from the incorporated existing knowledge and model-building assumptions. We highlight the microbial interaction prediction potential of cross-species co-expression analysis, which contrasts with these methods by its data-driven nature. We describe the utility of cross-species co-expression for various types of interactions and thereby inform future studies on use-cases of the approach and the opportunities and pitfalls that can be expected in its application.

**KEYWORDS** coexpression, metabolic gene cluster, antibiotics, synthetic community

Address correspondence to Robert A. Koetsier, robert.koetsier@wur.nl, Justin J. J. van der Hooft, justin.vanderhooft@wur.nl, or Marnix H. Medema, marnix.medema@wur.nl.

J.J.J.V.D.H. is a member of the Scientific Advisory Board of NAICONS Srl., Milao, Italy, and consults for Corteva Agriscience, Indianapolis, IN, USA. M.H.M. is a member of the Scientific Advisory Boards of Hexagon Bio and Hothouse Therapeutics Ltd.

See the funding table on p. 18.

Bacterial species in microbial communities engage in complex interaction networks that range from beneficial metabolic exchanges to competitive or exploitative dynamics. Understanding these interactions and identifying the genes, proteins, and metabolites involved therein is essential for unraveling the mechanisms that maintain community homeostasis or drive compositional changes (1). In various ecosystems, previous research has identified species that produce metabolites with a significant effect on the relative abundance and activity of other community members. For example, gut microbes produce metabolic by-products that create new niches for feeding, thereby enhancing community stability (2–4). Furthermore, certain rhizosphere microbes produce antibiotics that can eliminate plant pathogens or prevent their invasion (5). Identification of such interspecies interactions can help to inspire new treatment strategies for managing health, for example, in humans or agricultural ecosystems.

The most conclusive evidence for interspecies interactions is provided by experimental methods such as stable isotope labeling and microscopy combined with staining techniques; however, their implementation can be challenging and time-consuming (6). Since exhaustive experimental testing of all possible interactions is impractical, methods are needed to highlight the most plausible hypotheses for targeted investigations. This can be accomplished with a range of computational tools, varying in the datatype used to predict interactions between taxa. Driven by the increased popularity and availability of metabarcoding data, many tools popular in microbial ecology predict interspecies interactions based on co-occurrence. These include correlation-based approaches such as SparCC (7), CCLasso (8), SPIEC-EASI (9), and various ecological modeling approaches, e.g., based on the generalized Lotka–Volterra equation (reviewed in references 10, 11). A limitation of co-occurrence-based methods is that they do not provide hypotheses on the molecular mechanisms underlying interactions, which can be valuable for engineering functional communities or discovering drug candidates.

Modern systems biology approaches are better suited for generating hypotheses about the mechanisms underlying interspecies interactions. These approaches first reconstruct the metabolic networks of a community's taxa to describe all metabolite conversions encoded in their genomes. Following this, dynamic community flux balance analysis can be applied to predict the exchange rates of metabolites between microbial community members, or to predict competition (12). The construction of such models is complicated by the incomplete metabolic networks we have of most bacterial species and the need to select biologically relevant objective functions to accurately simulate microbial behavior under specific environmental conditions (12). A further limitation of models based on metabolic network reconstruction is that they are restricted to inferring interactions based on genes with pre-existing annotations. Crucially, it has been proposed that a large pool of unannotated genes could be involved in microbial interactions, possibly due to the common practice of studying gene functions in axenic cultures (13). This effect may be particularly pronounced for secondary metabolite biosynthetic gene clusters since their expression is often dependent on elicitors from the environment or microbial community members (14–17).

Recent studies have conducted co-expression studies with transcriptomics data from axenic cultures of various taxa to functionally classify previously unannotated genes (18–23). Extending these co-expression analyses to a cross-species context may offer a novel approach for exploring microbe-microbe interactions. While some previous studies have investigated cross-species co-expression, they have aimed to study host-microbe interactions (24–27), to dissect community functions into expression trait attributes (28), and to construct networks to determine community gene centrality (29, 30). To our knowledge, only two previous studies have utilized a cross-species co-expression approach to generate mechanistic hypotheses about microbe-microbe interactions occurring in co-cultures (31, 32). Those previous works demonstrated the hypothesis-generating potential of the method but did not yet systematically evaluate the findings in the context of experimentally characterized metabolic interactions. It would be desirable to further assess the predictions of cross-species co-expression using

multiple data sets that include strains with known reference information. Furthermore, the comparative strengths and weaknesses of the method in predicting different types of metabolic interactions—e.g., competition, cross-feeding, and specialized metabolic interactions—would need to be assessed.

Here, we aim to assess the feasibility of using cross-species co-expression analysis to predict various types of metabolic interactions, as well as to infer the functions of unannotated genes within microbial communities. For this, we use publicly available large-scale metatranscriptomics data sets from two synthetic microbial communities that were designed as model systems to study microbe-microbe interactions (14, 33). Our analyses revealed many putative cross-species associations. Several of these are likely to be spurious, for example, caused by pairs of microbial strains simultaneously responding to environmental changes rather than direct interactions. This is a potential pitfall of the method that should be addressed through careful experimental design, such as including (sufficient numbers of) conditions that specifically elicit responses in individual microbes. Nevertheless, other associations we assess likely represent real interactions. Specifically, our cross-species co-expression analysis successfully replicated findings from previous studies, such as the competition dynamics of the Mucin and Diet-based Minimal Microbiome (MDb-MM) community. Additionally, cross-species co-expression showed links between koreenceine BGC expression and expression of antibiotic (stress) response genes, suggesting that the method could provide a new strategy for inferring gene functions and for discovering novel antibiotics.

## RESULTS AND DISCUSSION

The approach adopted in this study focuses on the co-expression analysis of previously published RNA-sequencing data from bacterial co-cultures (13, 14, 33–36), Fig. 1A. The initial step of the cross-species co-expression workflow involves correcting for biases in expression levels caused by differences in sample library sizes and sample species' relative abundance. Next, we calculate the co-expression relationships between both genes and modules of functionally related genes. For module-centric analyses, we cluster genes based on their shared membership in metabolic pathways, as defined by antiSMASH (37) or the gut metabolic module (GMM) framework (38). To predict metabolic interactions from co-expression between microbes, we rely on the guilt-by-association principle, which assumes that co-expressed genes are related to the same biological process (20, 21), such as production and response to antibiotics, or nutrient sharing.

We conducted case studies on data from the Mucin and Diet-based Minimal Microbiome (MDb-MM) synthetic gut community to investigate primary metabolic interactions (33, 34), exploring both an untargeted network-based approach and conducting targeted investigations into competition and cross-feeding interactions (Fig. 1B). In addition, we analyzed expression data from the Hitchhikers of the Rhizosphere (THOR) synthetic community to explore specialized metabolic interactions (13, 14, 35) (Fig. 1B).

### Co-expression network analysis of GMMs reveals strong co-expression within species and between species with similar metabolic strategies

To investigate interspecies interactions in the Mucin and Diet-based Minimal Microbiome (MDb-MM) synthetic gut community, we represented metabolic pathways with the gut metabolic module (GMM) framework (38) and studied the associations between their expression. We initially adopted an untargeted approach, constructing a co-expression network by correlating GMM expression levels over time across three parallel bioreactors for all GMMs that passed filtering for low expression and coverage. We then explored whether biologically meaningful relationships could be extracted from the strongest edges of the co-expression network (Fig. 2A). Predominant among the strongest connections in the co-expression network were those linking metabolic processes of the same species, as can be seen by the clustering of nodes from the same bacterial species

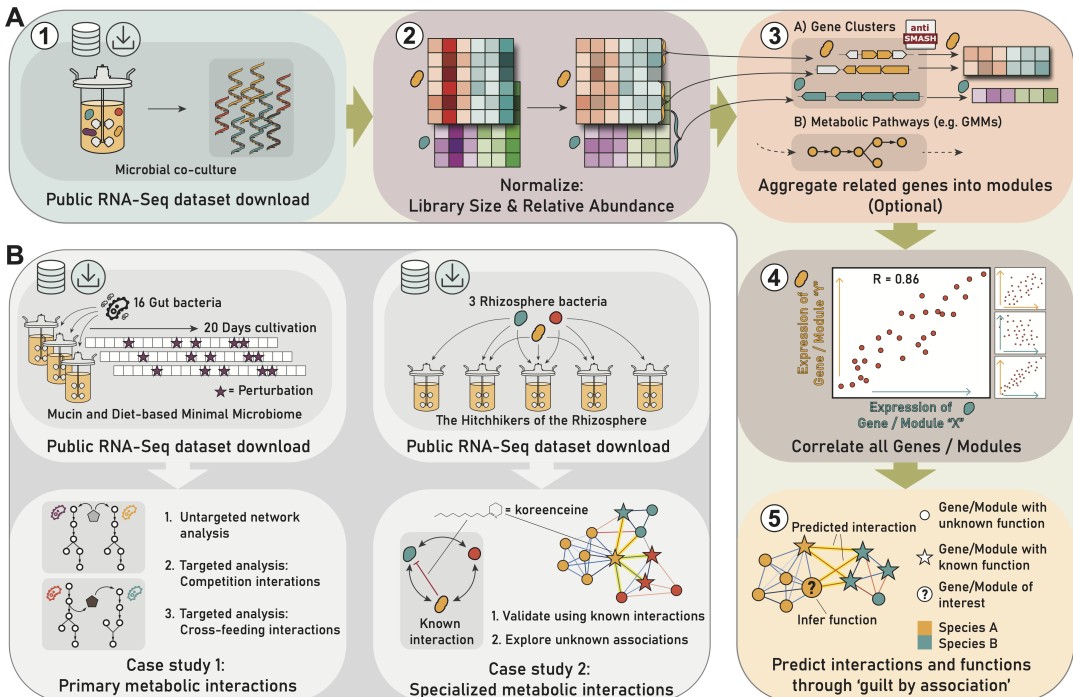

FIG 1  Overview of the workflow, data sets, and objectives of the conducted case studies. (A) General workflow; 1—RNA is isolated from various microbial community samples and sequenced so gene expression can be quantified for each microbe. 2—Gene expression data of each microbe is normalized per sample to account for differences in sample library size and microbial relative abundances. 3—Optionally, genes are grouped into modules. In this study, genes were grouped according to their membership in biosynthetic gene clusters (BGCs) detected by antiSMASH (37), and metabolic pathways defined by the gut metabolic module (GMM) framework (38). 4—Correlations between all genes (or gene modules) are quantified using Pearson's correlation coefficient. 5—The resulting co-expression relations are explored. Hypotheses can be generated for genes with unknown functions based on links to genes with known functions. Associations between genes' expression in different microbes may indicate that those genes play a role in microbial interactions. (B) Two datasets were analyzed in this study. The Mucin and Diet-based Minimal Microbiome (MDb-MM) (33) consists of 16 gut bacteria co-cultured in three parallel bioreactors, which were sampled longitudinally. Analyses of this microbial community focused on the co-expression of primary metabolic pathways. The second data set, the Hitchhikers of the Rhizosphere (THOR) (13, 14, 35), is composed of three rhizosphere bacteria cultured in various combinations, including some with a genetic mutant. The THOR community was analyzed to investigate associations between specialized metabolic pathways.

in Fig. 2A. Within individual species, in multiple cases, we identified strong co-expression between metabolic processes relating to carbohydrate degradation, energy generation, and amino acid biosynthesis (Table S1). For example, for *Akkermansia muciniphila*, we observed strong associations between the GMMs for fucose degradation and glycolysis ($r$ = 0.96) and arginine biosynthesis and glycolysis ($r$ = 0.98) (Fig. 2B). In other cases, we observed strong links between related metabolic pathways. For example, both for *Bacteroides ovatus* and *Faecalibacterium prausnitzii,* pectin degradation correlated with a pathway that utilizes a pectin breakdown product, galacturonate degradation ($r$ = 0.99 and $r$ = 0.96, respectively). Examples such as these suggest that the within-species edges in the co-expression network provide insight into their respective species' metabolic regulation and functionally related processes.

Particularly notable in the co-expression network in terms of cross-species relations are *Bacteroides xylanisolvens*, which is one of the predominant members of the MDb-MM, and *Bacteroides ovatus*, which is a member with relatively low abundance (33). For this species pair, we observed strong co-expression of many GMMs, which is reflected in the co-expression network through the clustering of nodes for these bacteria (Fig. 2A). We found that the metabolic processes co-expressed by these species are often identical. For instance, some of the most highly co-expressed GMMs of the *Bacteroides* pair correspond to their acetate synthesis ($r$ = 0.98), starch degradation ($r$ = 0.95), sucrose degradation ($r$ = 0.90), and alanine degradation ($r$ = 0.91), as illustrated in Fig. 2C. A more comprehensive analysis revealed that most shared GMMs of *B. xylanisolvens* and *B. ovatus* are

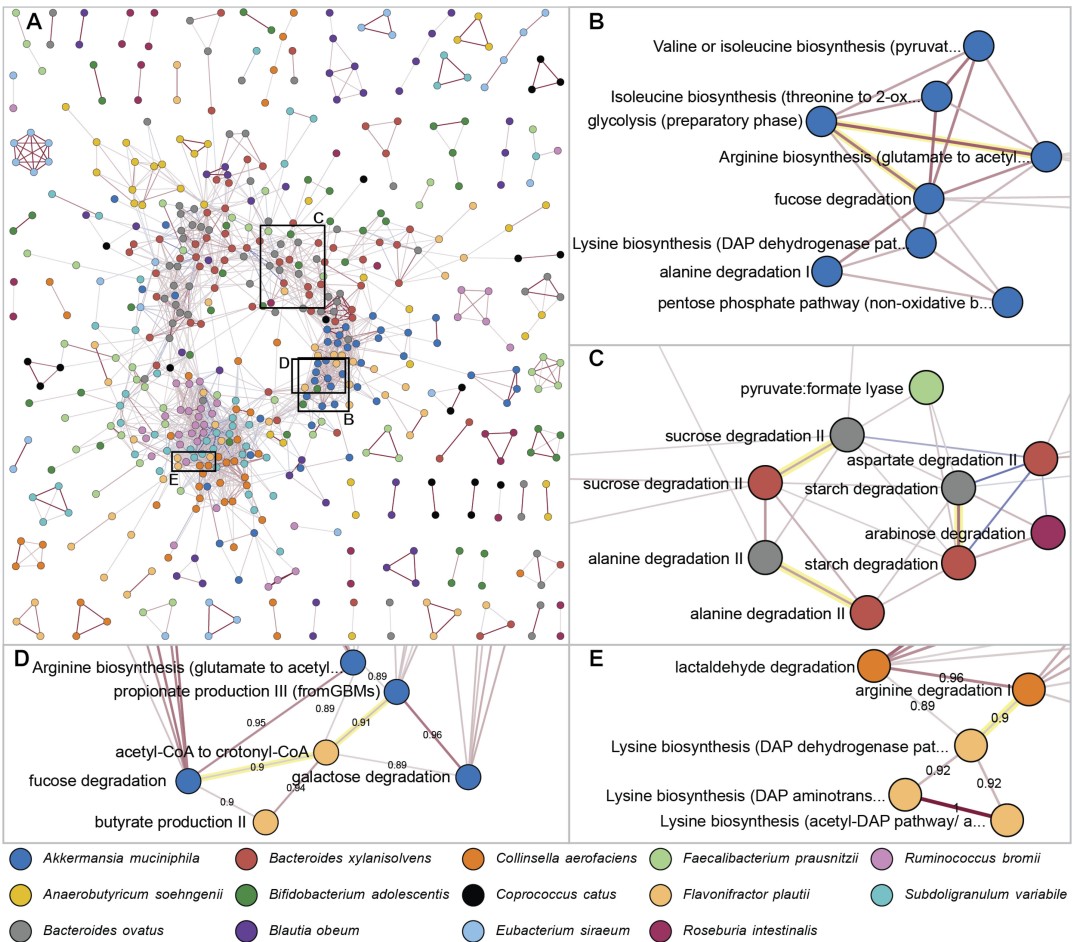

**FIG 2** Visualizations of the gut metabolic module (GMM) co-expression network. Nodes represent GMMs and are color-coded to indicate bacterial strains, and edges represent co-expression strength between GMMs based on Pearson's correlation coefficient, which has been scaled by the power of 6 while maintaining the sign for high contrast between strong edges. Network snapshots use different correlation cut-offs for sparser and more interpretable views. (A) Overview of the complete GMM co-expression network, singletons are excluded, and only edges with weight >0.2 (Pearson $r$ > 0.76) are shown. (B) Example of within-species co-expression, showing associations between *A. muciniphila* GMMs relating to substrate degradation, energy production, and amino acid biosynthesis. Only the strongest edges are shown, weight >0.5 ($r$ > 0.89). (C) Examples of GMMs that are highly co-expressed by *B. xylanisolvens* and *B. ovatus*, edge weights >0.3 ($r$ > 0.82). (D) and (E) show examples of strong interspecies edges, edge weights >0.5 ($r$ > 0.89). For a full and interactive version of the network, see 39 .

co-expressed, 76% (53 out of 70) of the GMMs exceed a correlation of 0.6 (Fig. S1). These results of cross-species co-expression align well with previous research, which calculated a high competitive index for *B. xylanisolvens* and *B. ovatus*, equally demonstrating similar activity of pathways for these species (34). Our results further illuminate the metabolic relationship between the *Bacteroides* species pair by providing a detailed view of pathways with similar or distinct expression dynamics. For example, dissimilar expression of the species' GMMs for allose degradation ($r = −0.31$) and arabinoxylan degradation ($r = −0.15$) indicates a possible nutritional complementarity that facilitates co-existence (Table S2).

Apart from the *Bacteroides* species, other species pairs also showed strong co-expression of metabolic pathways, though less frequently. Of all ~250 k cross-species correlation values, excluding those of the *Bacteroides* spp. pair, 0.16% (=398) surpasses the absolute value of 0.8. Not all these interspecies edges could be interpreted in the context of probable metabolic interactions, e.g., competition or cross-feeding. Among the most strongly co-expressed GMMs we found are *Flavonifractor plautii* acetyl-CoA to crotonyl-CoA vs *A. muciniphila* propionate production III ($r = 0.91$), *F. plautii* acetyl-CoA to crotonyl-CoA vs *A. muciniphila* fucose degradation ($r = 0.90$), *Collinsella aerofaciens*

arginine degradation vs *F. plautii* lysine biosynthesis ($r = 0.90$), and *Subdoligranulum variabile* valine or isoleucine biosynthesis vs *Ruminococcus bromii* pyruvate:ferredoxin oxidoreductase ($r = -0.90$); examples are shown in Fig. 2D and E, and more top edges can be found in Table S3. Given that these pathways are not closely connected in metabolic networks, bioreactor conditions—such as feeding cycles and shifts in nutrient profiles—likely confound the co-expression analysis by exerting a larger effect on species metabolism than direct microbial interactions. Consequently, cross-species co-expression primarily reflects distinct metabolic responses to bioreactor conditions rather than directly revealing microbial interactions (40). Nevertheless, these cross-species co-expression results agree with prior research that similarly identified distinct resource-utilization dynamics and suggested that these dynamics explain the community members' coexistence throughout the experiment (33).

Spurious associations caused by confounders are a common issue in correlation analyses (40), which can complicate interpretation in untargeted approaches. While the confounders are best minimized through careful experimental design, additional filtering of the interspecies edges based on likely pathway connections may help to prioritize the most probable microbial interactions. For example, one could focus exclusively on associations between pathways involved in potential cross-feeding or competition interactions, as we illustrate in the following sections.

## Cross-species co-expression of carbohydrate degradation GMMs recapitulates distinct resource utilization dynamics

We conducted a targeted exploration of resource utilization dynamics to examine how this relates to the co-existence of the MDb-MM community members. A design choice for the MDb-MM was to include bacteria that can utilize the same substrates, which can lead to competition for resources in the bioreactors (33). To assess whether cross-species co-expression can be effective at identifying such competition, we analyzed the associations between the expression of GMMs representing carbohydrate degradation pathways.

We initially explored the co-expression of experimentally confirmed carbohydrate degradation functions (Table S2 of reference 34). For example, it has been reported that fructose can be degraded by *C. catus*, *Eubacterium siraeum*, *F. plautii*, and *Roseburia intestinalis*, suggesting potential competition for this resource. In our analysis, *F. plautii* fructose degradation did not pass filtering, while the other species pairs showed low to moderate co-expression (Table 1). A similar trend is observed for species that are known to degrade starch: *Agathobacter rectalis*, *B. ovatus*, *E. siraeum*, *F. prausnitzii,* and *R. intestinalis* (34). *A. rectalis* could not be included due to low abundance throughout the experiment, while the other species again show low to moderate co-expression of the starch degradation module (Table 1).

For a more comprehensive view, we extended the analysis to include all carbohydrate degradation GMMs that were computationally inferred from the genomes of the MDb-MM. By calculating the co-expression between all pairs of equivalent GMMs from different microbes, we aimed to assess whether low to moderate similarity in expression profiles was unique to fructose and starch degradation or generally observed among carbohydrate degradation GMMs. Interestingly, our results suggested that low to moderate co-expression is the general trend that is observed for all carbohydrate degradation GMMs of the MDb-MM, as we detected few high correlations (>0.8) (Fig. 3). Nonetheless, the correlation distributions showed a slight positive bias, particularly for fructose, lactaldehyde, and mannose degradation. To test the significance of mean correlation shifts, we performed bootstrap resampling (50,000 iterations) for each carbohydrate degradation GMM. No significant negative shifts were detected. Significantly higher means than expected by chance were only found for fructose, lactaldehyde, and mannose degradation ($a = 0.05$, *P*-values: $2.2\times10^{-4}$, $8.0\times10^{-5}$, $4.2\times10^{-4}$, respectively), suggesting a relation in gene expression that is possibly driven by similar responses to environmental resource availability.

**TABLE 1** GMM pathway co-expression for known fructose and starch degraders

|  |  | *r* | Pathway |
|---|---|---|---|
| *C. catus* | - *E. siraeum* | 0.20 | Fructose degradation |
| *E. siraeum* | - *R. intestinalis* | 0.46 | Fructose degradation |
| *C. catus* | - *R. intestinalis* | 0.52 | Fructose degradation |
| *B. ovatus* | - *R. intestinalis* | −0.50 | Starch degradation |
| *F. prausnitzii* | - *R. intestinalis* | −0.50 | Starch degradation |
| *E. siraeum* | - *R. intestinalis* | −0.03 | Starch degradation |
| *E. siraeum* | - *F. prausnitzii* | 0.21 | Starch degradation |
| *B. ovatus* | - *E. siraeum* | 0.48 | Starch degradation |
| *B. ovatus* | - *F. prausnitzii* | 0.57 | Starch degradation |

Given that resource availability is identical for all bacteria residing in the same bioreactor, the overall limited bias toward higher positive correlations was initially unexpected. However, these observations align well with a previous study in which it was proposed that dynamic regulation of carbon source utilization pathways may facilitate coexistence among potential competitors in the MDb-MM by reducing metabolic overlap (33). Generally, transcriptional niche segregation has been suggested to play a crucial role in the co-existence of gut microbes (41). Our findings highlight that cross-species co-expression analysis can serve as an effective approach to identifying whether microbial pairs with comparable metabolic capabilities compete directly for specific resources, or whether transcriptional niche segregation reduces competition.

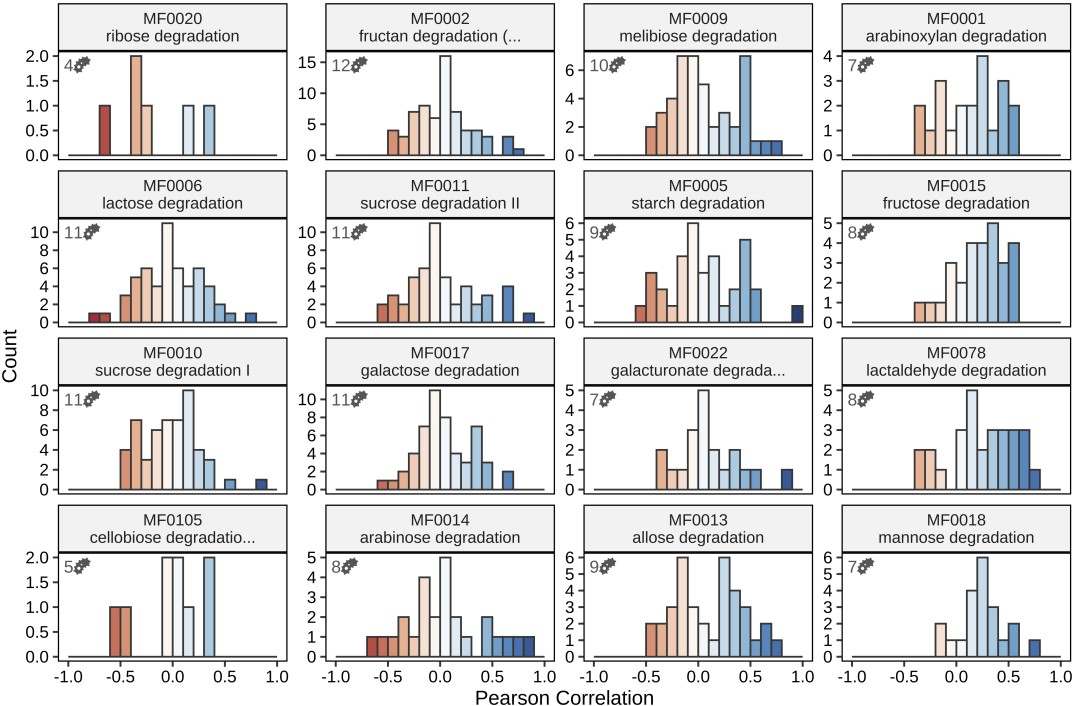

**FIG 3** Histogram summarizing the correlations observed between carbohydrate degradation GMM expression of different MDb-MM species. Most correlations are centered around zero, indicating generally distinct resource utilization patterns among organisms, with only a few extreme positive correlations. High co-expression (*r* > 0.8) was predominantly observed between the two bacteroides species, which shared four strongly co-expressed carbohydrate degradation pathways. A strong correlation (*r* = 0.82) was also detected between *Bifidobacterium adolescentis* and *F. prausnitzii*, specifically for sucrose degradation I (Table S2). Numbers in the top-left corner of each panel indicate the number of species for which this GMM was detected.

## Co-expressed metabolite production and consumption modules potentially highlight pairs of microbes for cross-feeding interactions

We assessed the potential of cross-species co-expression to prioritize microbial pairs in the MDb-MM that are likely to engage in cross-feeding interactions (Fig. 4A). We specifically focused on two cross-feeding interactions previously reported among bacteria of the MDb-MM community. First, we aimed to validate formate cross-feeding, known to occur in co-culture between *B. hydrogenotrophica* and *R. bromii*, the respective consumer and producer of this metabolite (42). Second, we focused on 1,2-propanediol, which is considered a key cross-fed metabolite in the MDb-MM community (33).

To investigate whether *B. hydrogenotrophica-R. bromii* formate cross-feeding occurs in the MDb-MM community, we conducted co-expression analysis of the formate-consuming Wood-Ljungdahl pathway (GMM: MF0097, "homoacetogenesis") and the formate-producing pyruvate formate-lyase (GMM: MF0074), which were proposed to be at the core of this interaction (42). The results of the co-expression between these pathways, for all MDb-MM species, are presented in Fig. 4B. Interestingly, a relatively low correlation is found between the pyruvate formate-lyase pathway of *R. bromii* and the Wood-Ljungdahl pathway of *B. hydrogenotrophica* ($r = 0.38$). The expression of *B. hydrogenotrophica's* formate-consuming pathway is instead most strongly correlated with the formate-producing pathway of *F. prausnitzii*. Notably, *F. prausnitzii* has been reported as the strongest formate producer in a study that considered a subset of the MDb-MM community (34). Since the Wood-Ljungdahl pathway is also present in *B. obeum*, it is interesting to observe whether this species mirrors the cross-feeding dynamics of *B. hydrogenotrophica*. Instead of correlating strongly with *F. prausnitzii*, we notice that the highest correlation for *B. obeum's* Wood-Ljungdahl pathway is its own pyruvate formate-lyase, indicating that *B. obeum* may consume self-produced formate. These results illustrate how cross-species co-expression analyses can be used to identify which cross-feeding interactions are most likely taking place, out of many possible pairs of species that could be interacting in a similar way.

Analysis of 1,2-propanediol cross-feeding yielded similar insights. We detected a GMM for 1,2-propanediol consumption in *Anaerobutyricum soehngenii* and *B. obeum*, and GMMs responsible for producing this compound were found in various species (Fig. 4C). Literature corroborates this metabolic capacity for several of these species; *A. soehngenii* and *B. obeum* are confirmed as consumers, and *A. muciniphila*, *B. ovatus*, and *B. obeum* as producers (33). The expression of the 1,2-propanediol consumption GMM of *A. soehngenii* matches most closely with the production modules of *B. ovatus* ($r = 0.59$) and *A. soehngenii* ($r = 0.57$), indicating that this species may consume its own 1,2-propanediol. *B. obeum's* consumption GMM is most strongly co-expressed with production GMMs of *F. prausnitzii* ($r = 0.68$) and *B. xylanisolvens* ($r = 0.62$) despite the existence of its own production module. The expression of the 1,2-propanediol producing GMM of *B. obeum* plummets after the first time point (when the bioreactors exit batch operation) and remains low (Fig. S2). The expression of its consumption module shows a peak at 48 h, after the first fasting period in the bioreactor. This peak in consumption matches best with a peak in production that is found at the same time point for *F. prausnitzii* and *B. xylanisolvens*, explaining the higher correlations observed in Fig. 4C. These findings suggest that *B. obeum* may benefit from 1,2-propanediol that is produced by these species during the first fasting period in the bioreactors. Again, this finding highlights the utility of cross-species co-expression analysis in identifying microbe pairs with the most compatible production and consumption dynamics. However, this analysis should be complemented by experimental validation since correlation-based methods alone do not confirm causal relationships.

Previous investigations have shown that microbes can express the same functions under different conditions (28, 41). While cross-species co-expression can be valuable to identify microbial pairs with compatible expression dynamics for cross-feeding, it is important to consider that expression levels do not necessarily correlate to metabolic flux (43), and co-expression is naïve to the scale of metabolic conversions. Scale issues

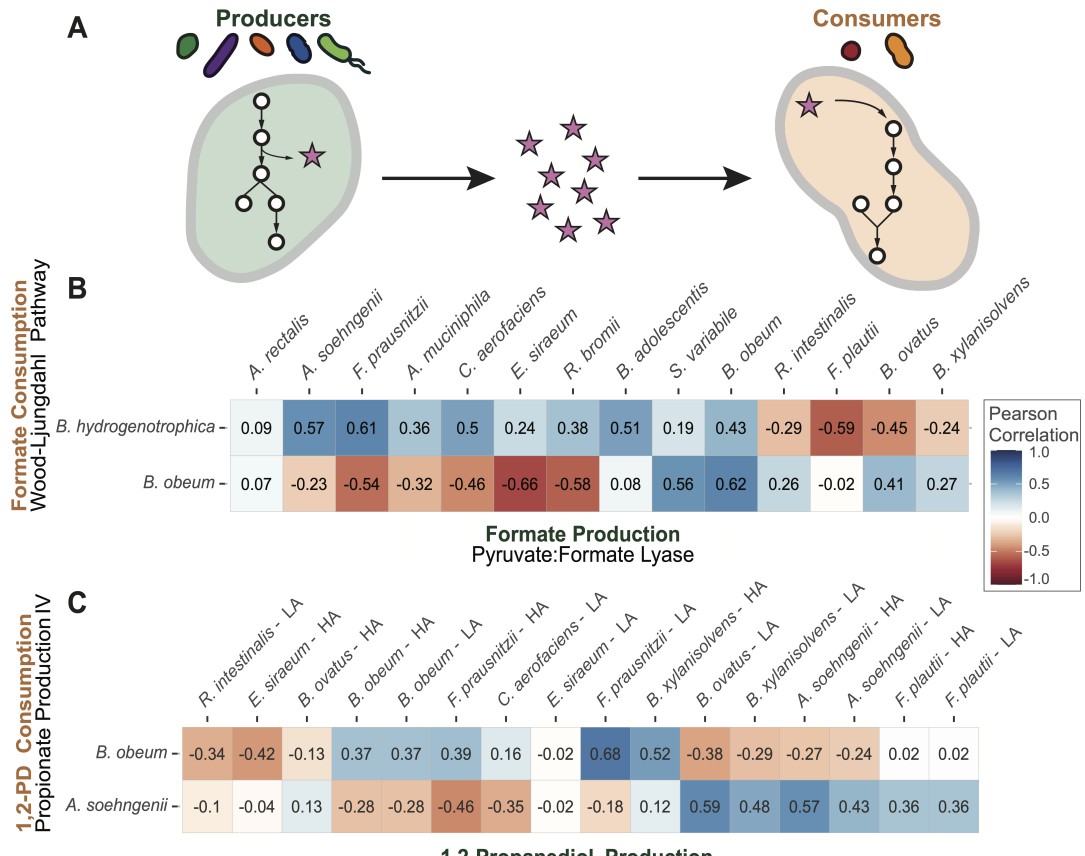

**FIG 4** Cross-feeding microbial pairs in the MDb-MM are predicted with cross-species co-expression. (A) Illustration of cross-feeding interactions. Producers (left) release metabolites into the environment, which can be utilized by other community members (right). By studying the co-expression of the depicted pathways, we aimed to detect pairs of microbes with similar production and consumption dynamics, thus increasing the chance that a cross-feeding relationship exists between the species. (B) and (C) are correlation heatmaps showing co-expression between GMMs involved in metabolite production (columns) and consumption (rows), for formate and 1,2-propanediol, respectively. For 1,2-propanediol production, either lactaldehyde (LA) or hydroxyacetone (HA) may be used as a substrate. GMMs are computationally inferred; not all the depicted pathways have experimental evidence confirming the presence of the corresponding functions in each species.

may arise when a low-abundance community member with high pathway expression produces smaller quantities of a metabolite than a high-abundance member with relatively lower expression. This challenge complicates the disentanglement of species' relative contributions when investigating redundant functions, a common occurrence in gut microbial communities (38, 44). Hence, the method may prove most powerful when microbial interactions relying on non-redundant functions are investigated. Specialized metabolic pathways can be interesting targets given their high diversity at the taxonomic level of families or higher (45). We explore this in the following section.

## Cross-species co-expression analysis suggests cross-talk in specialized metabolic pathways producing koreenceine, siderophores, and bacillithiol

To investigate the efficacy of cross-species co-expression for identifying interactions on the level of secondary metabolism, we conducted a case study on the Hitchhikers of the Rhizosphere (THOR) community. We focused on secondary metabolic pathways that were predicted by the genome mining tool antiSMASH (37), following the example of Chevrette et al. (14), to use their study as a benchmark for evaluating our results. antiSMASH predicted 44 biosynthetic gene cluster (BGC) regions for *B. cereus*, 28 for *F. johnsoniae*, and 32 for *P. koreensis*. Among these were BGCs that have been characterized in previous research, for example, those responsible for the synthesis of siderophores

petrobactin and bacillibactin for *B. cereus*, antioxidant flexirubin for *F. johnsoniae*, and antibiotic koreenceine for *P. koreensis* (as listed in reference 14). After pre-processing the BGCs by filtering for low expression and refining the boundaries, 23, 1, and 7 BGCs were removed, and 3, 5, and 2 BGCs were split into parts, for *B. cereus*, *F. johnsoniae*, and *P. koreensis*, respectively (Table S4 and Supplemental File 1).

We analyzed the co-expression of the remaining BGCs between species pairs to detect interactions between the secondary metabolic pathways. Most notable among the results was the high co-expression (either positive or negative) between the BGC for the antibiotic koreenceine (*P. koreensis*), and several BGCs of the other two species (Fig. 5A; Fig. S4). Particularly striking was the strong negative association between the expression of the koreenceine BGC and siderophore BGCs of all three THOR community species. Namely, strong negative correlations were found with *B. cereus* BGCs for petrobactin ($r = -0.92$) and bacillibactin ($r = -0.87$), and the *F. johnsoniae* BGC for a fulvivirgamide-like siderophore ($r = -0.91$). Even the *P. koreensis* BGCs for a pyoverdine-like siderophore showed anti-correlation with koreenceine expression ($r = -0.76$ for the peptide backbone locus and $r = -0.60$ for the chromophore nonribosomal peptide synthetase [NRPS] locus, Fig. S6). These results extend the findings of prior work that associated koreenceine and siderophore expression only for *B. cereus* (14), but our further observations indicate that this relationship could be indirect. Deleting koreenceine in *P. koreensis* increases expression of its pyoverdine-like siderophore BGC in axenic culture, and the same shift occurs in co-culture (Fig. 5B). Thus, altered *P. koreensis* siderophore expression, rather than koreenceine itself, may drive siderophore induction in other community members.

Among the BGCs exhibiting a strong positive correlation with koreenceine expression, we detected bacillithiol synthesis loci 1 and 2 of *B. cereus* ($r = 0.86$ and $r = 0.83$, respectively). The co-expressed core of antiSMASH BGC 1.8 a, which partially overlaps with bacillithiol synthesis locus 3, did not include the *bshC*, the gene crucial for the final step of bacillithiol synthesis (46). While the co-expression of BGC 1.8a with the koreenceine BGC is low, we found a high correlation for the *bshC* gene, $r = 0.92$, confirming high co-expression with the koreenceine BGC for all crucial bacillithiol biosynthetic genes. This may again represent an indirect association, given the small expression difference of bacillithiol in axenic *B. cereus* culture and its co-culture with wild-type *P. koreensis* (Fig. 5B). Instead, iron availability possibly drives bacillithiol expression changes. Iron availability for *B. cereus* could increase when the *P. koreensis* koreenceine BGC deletion mutant, which produces elevated siderophore levels, is absent from the culture. This could lead to increased levels of free $Fe^{2+}$ in *B. cereus*, exacerbating oxidative stress through the Fenton reaction (47). Bacillithiol might mitigate this stress, given its known role in maintaining redox balance and detoxifying active oxygen species (48, 49). An alternative hypothetical explanation could be that the antimicrobial activity of koreenceine leads to oxidative stress in *B. cereus*, and bacillithiol may help counter this.

By highlighting the anti-correlation between expression of koreenceine and the pyoverdine-like siderophore in *P. koreensis*, cross-species co-expression analysis allowed us to develop more refined hypotheses regarding the interactions underlying changes in iron acquisition dynamics within the community. These results demonstrate how cross-species co-expression can be leveraged to generate valuable functional hypotheses about gene interactions influencing community ecology.

## Cross-species co-expression shows potential for identifying signatures of antibiotic response

The production of the antibiotic koreenceine has a significant impact on gene expression in the THOR community (13, 50). We conducted a targeted analysis of the genes co-expressed with koreenceine to determine whether cross-species co-expression could reveal its antibiotic role and provide insights into its mechanism of action.

For this targeted analysis, we examined the association of koreenceine BGC expression with any cluster of co-localized and co-expressed genes in *B. cereus* and *F.*

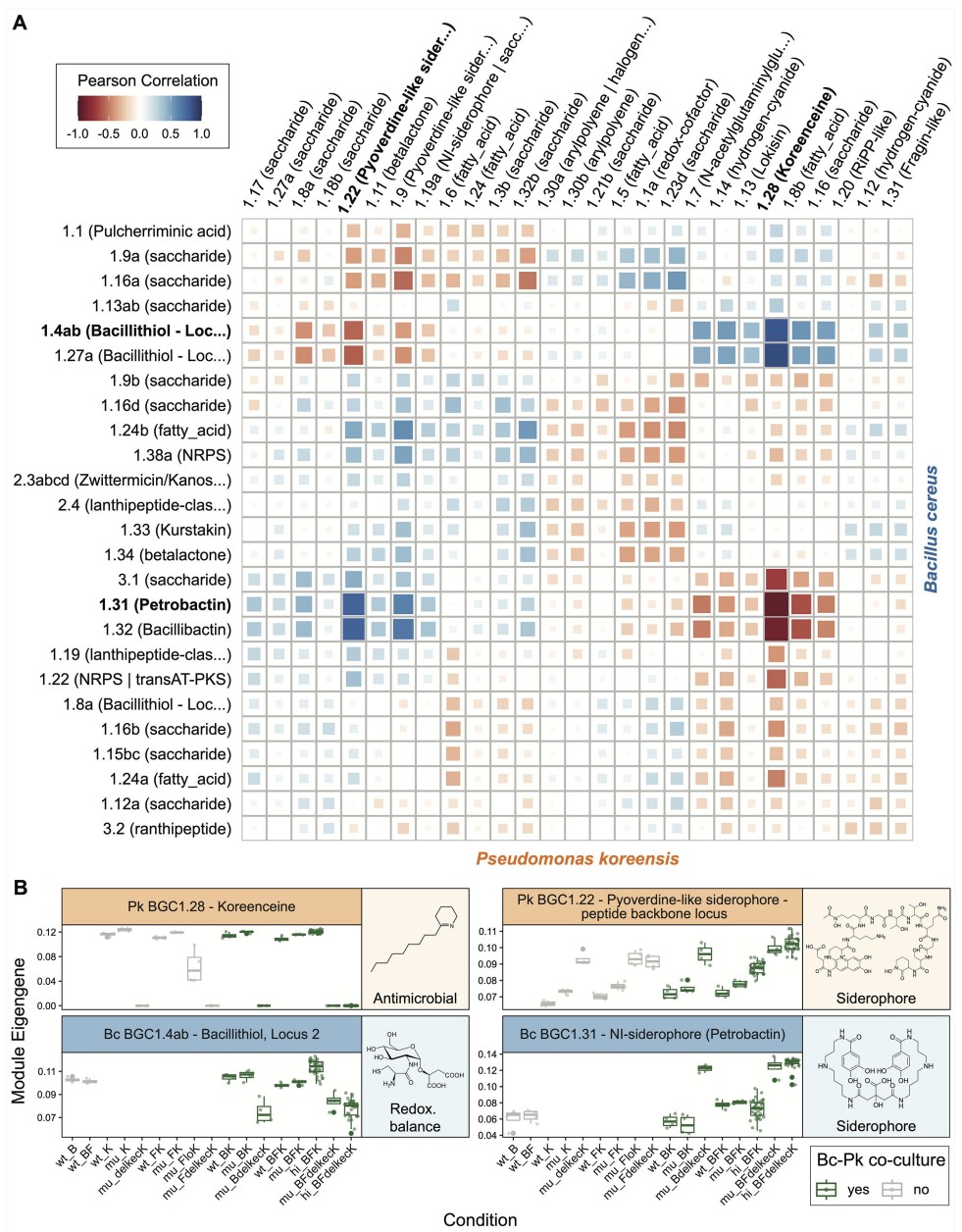

**FIG 5** The koreenceine BGC (*P. koreensis*) exhibits strong co-expression with BGCs of the other THOR community members. (A) Heatmap showing co-expression between all *P. koreensis* and *B. cereus* BGCs. Siderophores are strongly anti-correlated with koreenceine, while the bacillithiol synthesis loci show a strong positive correlation. See Fig. S3-S5 for BGC-BGC correlations of all THOR species pairs. (B) Expression profiles of koreenceine and strongly correlating BGCs from *B. cereus* (see Fig. S7 for more BGC expression profiles). The *x*-axis labels describe different community compositions, B = *B. cereus*, F = *F. johnsoniae*, K = *P. koreensis*, while prefixes describe different experiments, wt = wild type, no mutants were included in this round of experiments, mu = mutant, this experimental round introduced some conditions with *P. koreensis* koreenceine deletion mutants (delkecK) and a low inoculum condition (loK), hi = high replicate, this experimental round was characterized by conditions with very high replicate numbers. BGC expression is visualized with the cluster's eigengene, which summarizes the expression of the BGC's genes by taking the first principal component of their scaled and log-transformed counts. Note that co-expression was calculated across samples and not conditions, as shown here.

*johnsoniae*, extending beyond the BGC-limited scope of the previous analysis. To guide functional interpretation, we reviewed the co-expressed genes' putative functions as assigned by eggNOG-mapper (51). Several genes involved with responses to antibiotics

were detected among the most strongly co-expressed genes (Table 2). For example, we found two RND multidrug efflux systems co-located and co-expressed with *tetR* transcriptional regulators of *F. johnsoniae*. Supporting the notion that the expression of these RND genes is affected by antibiotics, a previous study on *F. johnsoniae* found one of the RND clusters upregulated in response to the antibiotic chloramphenicol (52).

In addition, we found various reductase-encoding genes and other loci linked to oxidative or disulfide stress responses that are strongly co-expressed with the koreenceine BGC (Table 2). Among the reductase-related genes, we detected the flavodoxin-2-domain containing *azoR* (both *B. cereus* & *F. johnsoniae*) and *ywqN* (*B. cereus*). Genes associated with oxidative-stress responses included *bshC* (*B. cereus*, mentioned previously), *msrA* (both *B. cereus* & *F. johnsoniae*)—encoding a methionine-sulfoxide reductase that repairs proteins by reducing oxidized methionine residues (53)—and a member of the selO family, which has previously been associated with oxidative stress response (54). Interestingly, *spxA2* (*B. cereus*), which encodes a global transcriptional regulator that responds to oxidative and disulfide stress, was also found to be highly co-expressed. This transcription factor may regulate some of the genes co-expressed with the koreenceine BGC since its regulon includes various genes involved with responding to oxidative stress and maintaining thiol homeostasis, among which bacillithiol (55, 56).

Among the anti-correlated genes, we found that those related to iron acquisition or oxidative processes were prevalent (Table 2). The iron acquisition set included genes encoding siderophore biosynthetic enzymes, as discussed previously, transporters that are frequently involved in iron uptake—e.g., TonB-dependent transporters (57)—and the heme-degrading iron-releasing enzyme IsdG (*B. cereus*) (58). As before, the expression of these iron-acquisition genes increased when strains were co-cultured with the siderophore-producing *P. koreensis* koreenceine deletion mutant, whereas co-culture with wild-type *P. koreensis* had little effect (Fig. S8). This pattern supports the idea that siderophore production by *P. koreensis*, rather than koreenceine itself, drives community-wide shifts in iron metabolism. In contrast, the anti-correlated genes linked to oxidative processes appeared to be more directly associated with koreenceine expression. For instance, the genes encoding a $cbb_3$type cytochrome $c$ oxidase and a 2-nitropropane dioxygenase (*F. johnsoniae*) were strongly downregulated in co-culture with wild-type *P. koreensis* (which expresses koreenceine) but not in any other conditions (Fig. S8). This observation again suggests that koreenceine can affect redox balance.

To summarize, our co-expression results reflect koreenceine's role as an antibiotic, and its association with redox- or disulfide stress and may provide insights into the molecular means by which this specialized metabolite influences the THOR community ecology. Further insights may be gained from the link of koreenceine with the SpxA2 regulator. Specifically, the factors influencing SpxA2 activity, such as oxidative stress and potentially cell wall or membrane stress (56, 59, 60), may point toward the mechanism through which koreenceine affects microbes.

## Conclusions and future perspectives

In this work, we established whether microbe-microbe interactions can be predicted based on associations between gene expression across different species. Our untargeted cross-species co-expression analysis revealed some spurious associations, while the results of the targeted approach effectively recapitulated previously described insights into microbial nutrient utilization dynamics. Furthermore, the method showed a potential to highlight pairs of microbes with compatible expression dynamics for cross-feeding. Finally, we confirmed that antibiotic response genes are co-expressed with the koreenceine BGC, consistent with the known biological role of this metabolite. These findings confirm that cross-species co-expression can be used to prioritize hypotheses of microbe-microbe interactions, which can be experimentally validated.

Compared to existing approaches for studying microbial interactions, such as methods analyzing co-occurrence, cross-species co-expression analysis distinguishes

**TABLE 2** Genes from *B. cereus* (Bc) and *F. johnsoniae* (Fj) that are co-expressed with the koreenceine BGC of *P. koreensis*[a]

| Co-expr. locus nr. | Category | Org. | Locus tag | Prodigal tag | Gene symbol | PCC | Description | PFAM domain |
|---|---|---|---|---|---|---|---|---|
| 676 | Reduction | Bc | J3D61_RS10655 | ctg1_2119 | ywqN | 0.97 | NAD(P)H-dependent | Flavodoxin_2 |
| 1,719 | Antibiotic response | Bc | J3D61_RS29505 | ctg2_344 | | 0.93 | Metallo-beta-lactamase superfamily | Lactamase_B |
| 1,670 | Antibiotic response | Bc | J3D61_RS28235 | ctg2_82 | | 0.93 | Metallo-beta-lactamase superfamily | Lactamase_B |
| 1,421 | Reduction | Bc | J3D61_RS23250 | ctg1_4533 | azoR | 0.92 | Catalyzes the reductive cleavage of azo bond in aromatic azo compounds to the cor... | Flavodoxin_2 |
| 240 | Ox. stress response | Bc | J3D61_RS04420 | ctg1_895 | bshC | 0.92 | Involved in bacillithiol (BSH) biosynthesis. May catalyze the last step of the pa... | BshC |
| 765 | Ox. stress response | Bc | J3D61_RS12025 | ctg1_2396 | msrA | 0.9 | Has an important function as a repair enzyme for proteins that have been inactiva... | PMSR,SelR |
| 80 | Ox. and disulfide stress response | Bc | J3D61_RS01450 | ctg1_294 | spxA2 | 0.9 | Interferes with activator-stimulated transcription by interaction with ... | ArsC |
| 468 | Iron acquisition | Bc | J3D61_RS07510 | ctg1_1514 | isdG | −0.88 | Allows bacterial pathogens to use the host heme as an iron source. Catalyzes the ... | ABM |
| 1,532 | Reduction | Fj | FJOH_RS23410 | ctg1_4657 | | 0.99 | NADPH-quinone reductase | Flavodoxin_2 |
| 1,176 | Reduction | Fj | FJOH_RS17215 | ctg1_3404 | azoR | 0.98 | Catalyzes the reductive cleavage of azo bond in aromatic azo compounds to the cor... | Flavodoxin_2 |
| 1,033 | Ox. stress response | Fj | FJOH_RS14495 | ctg1_2861 | | 0.97 | Belongs to the UPF0061 (SELO) family | UPF0061 |
| 1,033 | Ox. stress response | Fj | FJOH_RS14500 | ctg1_2862 | msrA | 0.97 | Has an important function as a repair enzyme for proteins that have been inactiva... | PMSR |
| 438 | Antibiotic response | Fj | FJOH_RS06575 | ctg1_1300 | | 0.93 | tetR family | TetR_N |
| 438 | Antibiotic response | Fj | FJOH_RS06580 | ctg1_1301 | | 0.93 | PFAM Outer membrane efflux protein | OEP |
| 438 | Antibiotic response | Fj | FJOH_RS06585 | ctg1_1302 | | 0.93 | Belongs to the membrane fusion protein (MFP) (TC 8.A.1) family | HlyD_D23 |
| 438 | Antibiotic response | Fj | FJOH_RS06590 | ctg1_1303 | | 0.93 | Belongs to the resistance-nodulation–cell division (RND) (TC 2.A.6) family | ACR_tran |
| 1,470 | Antibiotic response | Fj | FJOH_RS22225 | ctg1_4412 | | 0.91 | TetR family transcriptional regulator | TetR_N, WHG |
| 1,470 | Antibiotic response | Fj | FJOH_RS22230 | ctg1_4414 | | 0.91 | Belongs to the membrane fusion protein (MFP) (TC 8.A.1) family | Biotin_lipoyl_2, HlyD_3, HlyD_D23 |
| 1,470 | Antibiotic response | Fj | FJOH_RS22235 | ctg1_4415 | | 0.91 | Belongs to the resistance-nodulation–cell division (RND) (TC 2.A.6) family | ACR_tran |
| 1,470 | Antibiotic response | Fj | FJOH_RS22240 | ctg1_4416 | | 0.91 | RND transporter | OEP |
| 1,175 | Iron acquisition | Fj | FJOH_RS17205 | ctg1_3402 | | −0.86 | Tonb-dependent siderophore receptor | CarbopepD_reg_2, Plug, TonB_dep_Rec |
| 1,003 | Iron acquisition | Fj | FJOH_RS14055 | ctg1_2778 | | −0.87 | TonB-dependent receptor | CarbopepD_reg_2, Plug, TonB_dep_Rec |
| 61 | Iron acquisition | Fj | FJOH_RS00820 | ctg1_153 | | −0.87 | Iron dicitrate transport regulator FecR | DUF4974, FecR |
| 1,465 | Iron acquisition | Fj | FJOH_RS22075 | ctg1_4384 | | −0.89 | TonB-dependent Receptor Plug | CarbopepD_reg_2, Plug, TonB_dep_Rec |
| 1,129 | Iron acquisition | Fj | FJOH_RS16490 | ctg1_3260 | | −0.91 | TonB-dependent Receptor Plug | CarbopepD_reg_2, CarboxypepD_reg, Plug, TonB_dep_Rec |
| 1,129 | Iron acquisition | Fj | FJOH_RS16455 | ctg1_3253 | | −0.91 | Siderophore biosynthesis protein domain | Acetyltransf_8, FhuF, lucA_lucC |
| 945 | Oxidation | Fj | FJOH_RS13160 | ctg1_2599 | ccoS | −0.93 | Cytochrome oxidase maturation protein | FixS |
| 945 | Oxidation | Fj | FJOH_RS13165 | ctg1_2600 | ccoN | −0.93 | Belongs to the heme-copper respiratory oxidase family | COX1, FixO |
| 945 | Oxidation | Fj | FJOH_RS13170 | ctg1_2601 | ccoQ | −0.93 | Cytochrome c oxidase subunit IV | FixQ |

*(Continued on next page)*

**TABLE 2** Genes from *B. cereus* (Bc) and *F. johnsoniae* (Fj) that are co-expressed with the koreenceine BGC of *P. koreensis*[a] (*Continued*)

| Co-expr. locus nr. | Category | Org. | Locus tag | Prodigal tag | Gene symbol | PCC | Description | PFAM domain |
|---|---|---|---|---|---|---|---|---|
| 945 | Oxidation | Fj | FJOH_RS13175 | ctg1_2602 | *ccoP* | −0.93 | Cytochrome *c* oxidase | Cytochrome_CBB3, FixP_N |
| 945 | Oxidation | Fj | FJOH_RS13180 | ctg1_2603 | *ccoG* | −0.93 | Cytochrome *c* oxidase | Fer4_18, Fer4_5, FixG_C |
| 945 | Oxidation | Fj | FJOH_RS13185 | ctg1_2604 | *ccoH* | −0.93 | Cytochrome cbb3 oxidase maturation protein CcoH | FixH |
| 945 | Oxidation | Fj | FJOH_RS13195 | ctg1_2606 | *hemN* | −0.93 | Belongs to the anaerobic coproporphyrinogen-III oxidase family | HemN_C,Radical_SAM |
| 164 | Oxidation | Fj | FJOH_RS02665 | ctg1_518 | | −0.93 | 2-Nitropropane dioxygenase | |

[a]Only highly correlated genes linked to reduction processes, iron metabolism, or responses to oxidative stress and antibiotics are shown. The columns represent: co-expressed locus number, an index for groups of co-located and co-expressed genes; Category, manually assigned categories based on gene descriptions; Org., the gene-carrying organism; Locus tag, gene identifiers used in the NCBI genome annotations; Prodigal tag, gene identifiers used internally for the analyses; Gene Symbol, derived from the eggNOG output; PCC, Pearson's correlation coefficient, quantifying association with koreenceine BGC expression; Description, functional annotation from eggNOG; PFAM Domain, identified by eggNOG. The complete list is available in Table S5 and S6; expression profiles of the listed genes are shown in Fig. S8.

itself with its unique ability to provide insight into the pathways that may underlie the interactions. Furthermore, the method stands out for its simplicity of implementation. Unlike other methods for predicting microbe-microbe interactions, such as flux balance analysis-based methods (12), cross-species co-expression studies do not require complex parameterization or the selection of objective functions. A concomitant downside is that the correlation-based approach does not predict metabolite quantities, making it hard to assess the physiological relevance of observed associations, for example, potential cross-feeding interactions. Hence, some care should be taken when interpreting strong associations in gene expression between microbes as interactions since these associations may similarly arise from confounding factors.

Careful selection of conditions included in the experimental design can help to reduce confounders. Variation of environmental factors such as temperature, pH, and general nutrient availability can lead to indirect associations as they affect many species simultaneously (61). Including experimental conditions that are likely to elicit gene expression changes in a single microbe—such as genetic modification or variation of highly specific nutrients—can be advantageous to distinguish experimental treatment effects from interaction effects. An alternative approach, more suitable for scaling to larger communities, is the drop-out experimental design (62, 63), which can help to identify co-expressed genes specifically responsive to the absence of individual organisms. Associations between gene expression levels are an indirect way of detecting microbial interactions, as it is often the gene's downstream metabolite products that mediate interactions (64). Since this indirectness can introduce additional noise, we advise prioritizing experimental conditions expected to elicit strong, binary-like (on/off) transcriptional responses in the expression of pathways of interest. Focusing investigations on secondary metabolite biosynthetic gene clusters can be beneficial since they remain cryptic in many conditions, being activated only by specific elicitors (15–17). In addition to selecting appropriate experimental conditions, acquiring sufficient data is essential for powerful co-expression analyses. Sequencing depth should be adjusted based on the community's size and evenness of relative abundances, and including more than 20 samples has been recommended for co-expression studies (65, 66).

The efficacy of cross-species co-expression could be further improved through additional methodological development. The approach presented here relies on co-culture samples to infer both intra- and interspecies gene associations in the co-expression network. Creating a hybrid network by integrating additional (public) data sets to achieve more accurate predictions of intraspecies co-expression may help disentangle interaction mechanisms. For example, information on independently modulated gene sets (67, 68) could assist in separating an organism's multifactorial responses—such as those to iron starvation and antibiotics—that occur simultaneously in a given co-culture experiment.

In summary, we have demonstrated that cross-species co-expression successfully replicates known microbial interaction patterns, confirming the method's feasibility for predicting microbial interactions. The association of koreenceine BGC expression with antibiotic response genes illustrates that the method can support prioritization or identification of gene targets in antimicrobial discovery efforts. Beyond its use in natural product research, the method may facilitate microbial ecology studies. While cross-species co-expression analysis cannot establish causal functional links, it can reduce the experimental labor involved in mechanistic studies by generating hypotheses about gene sets relevant to microbe-microbe interactions. For example, in microbial communities with observed interspecies inhibition interactions, candidate BGCs potentially involved in chemical warfare between microbiome members (e.g., through antibiotic action) could be prioritized based on co-expression with stress response genes, such as those encoding efflux pumps. Through such applications, cross-species co-expression analysis can help understand shifts in microbial communities, including those underlying disease development.

## MATERIALS AND METHODS

### Genomes, BGC prediction, and filtering, functional annotation—THOR analyses

Genomic sequences for the THOR community species were obtained from the GitHub repository associated with (14), which provides access to the following genomes: *Flavobacterium johnsoniae* strain UW101 (NCBI GenBank assembly GCA_000016645.1), *Bacillus cereus* strain UW85, and *Pseudomonas koreensis* strain CI12 (JGI-IMG/M accessions Ga0417192 and Ga0417193, respectively).

BGCs were identified using antiSMASH (v7.1.0) with "loose" detection strictness to ensure comparability with the analyses conducted in a previous study (14). By default, antiSMASH determines BGC boundaries based on a fixed offset defined by its cluster detection rules and may, thus, include genes unrelated to the biosynthetic process in a BGC (37, 69). To improve cluster boundaries, the genes inside a BGC were separated into clusters of co-located (maximum intergenic distance of 300 bp for each pair) and co-expressed ($r > 0.5$) genes. As a result of this step, some BGCs may be divided into multiple sub-clusters that are identified by suffixes (e.g., BGC_1.9a and BGC_1.9b). Each sub-cluster must contain at least one core biosynthetic gene to avoid being discarded. In some instances, experimental evidence on BGC gene composition (Table S2 of reference 14) allowed for further refinement of BGC boundaries. The refined clusters were subjected to a filtering step, removing BGCs that had no core biosynthetic gene passing the low expression threshold. Finally, additional filtering was performed through manual curation, excluding BGCs where the majority of the genes (>70%) did not pass the low expression filter or where internal co-expression was very poor (no $r > 0.5$ for core genes). See Table S4 for the full list of BGCs and which filtering criteria apply and Supplemental File 1 for the final BGC boundaries.

Functional prediction was conducted for proteins identified in the GenBank output generated by antiSMASH, which utilized Prodigal (v2.6.3) (70) to predict protein-coding genes. The extracted protein sequences were then functionally annotated using eggNOG-mapper (v2.1.12) (51, 71).

### RNA-seq data acquisition and processing—MDb-MM and THOR analyses

RNA-seq count data for the Mucin and Diet-based Minimal Microbiome (MDb-MM) synthetic community were obtained from the 'syncomR' R package (available at https://github.com/microsud/syncomR) (33). For analyses conducted on the THOR community, we utilized read count data provided by Marc G. Chevrette and colleagues, generated with the methodology described in (14). The sequences from which these data are derived are stored in NCBI's Sequence Read Archive (SRA) under the BioProject accession number PRJNA885088 and PRJNA1321520 (13, 14, 36).

In preparation for co-expression analysis, filtering was applied to remove low-expression genes, retaining only those with at least 10 read counts in at least half of the samples. The remaining gene expression counts were normalized for each taxon individually (72), using the trimmed mean of M-values from 'edgeR' v3.42.2 (73). This approach accounts for changes in the relative abundance of the species between samples, removing the strong positive bias observed with correlations between raw read counts (Fig. S9 and S10). Subsequently, the normalized expression values were transformed using a hyperbolic arcsine transformation (74).

### Gut metabolic modules—MDb-MM analyses

While available in the 'syncomR' package, the expression of gut metabolic modules (GMM) was recalculated to incorporate the data preparation steps we selected for co-expression analysis. GMM expression was calculated with the R package omixerRpm v0.3.3 (available at https://github.com/omixer/omixer-rpmR) (75). As an input for this step, we utilized gene expression data that had been filtered, normalized, and annotated

with KEGG terms from the 'syncomR::SynComRNAKEGG'. Before running omixerRpm, genes contributing to the same KO terms were summed, and a hyperbolic arcsine transformation was applied on the KO term level. OmixerRpm was run with minimum.coverage = 0.5, score.estimator = "median", annotation = 2 to differentiate between taxa, and the database file specifying GMM KO term composition curated by Shetty et al. (33).

## Cross-species co-expression analysis—MDb-MM and THOR analyses

Correlations between pairs of genes or pairs of GMMs were quantified using the Pearson correlation coefficient, computed with the cor function of WGCNA v1.72 in R v4.3.0 (76, 77). For cross-species correlation, only samples containing both species were included in the calculations. Some GMM pairs correlated strongly due to outlier samples. To account for this, any edge between a GMM pair exhibiting a high discrepancy between its Pearson and Spearman correlation (>0.3) was removed.

For MDb-MM network analysis, the correlations between GMMs were scaled using a power function while maintaining the sign of the correlation. A power of 6 was chosen for scaling based on the fit to a scale-free topology model, a procedure that is common in WGCNA analysis (76). The network was visualized in Cytoscape v3.10.0, and visual styling was applied with the R package RCy3 v2.20.0 (78, 79). Within-species edges in the network may exist between GMMs that are partially composed of the same KEGG terms. These edges were excluded from interpretations listed in this manuscript, as their overlap artificially inflates correlation coefficients. Table S1 lists the strongest within-species edges along with the quantity of shared KEGG terms expressed as a fraction of the total unique KEGG terms that compose the GMMs.

For the analysis of interactions in THOR secondary metabolism, associations were determined between groups of genes that were co-located in the genome and co-expressed. This framework was applied both to BGCs and non-BGC clustering genes. Co-expression between gene clusters was calculated by averaging all inter-cluster gene-gene Pearson correlation coefficients. To reduce bias in this step, correlation values are Fisher z-transformed before averaging, followed by inverse-transforming the mean back to the original scale (80).

## ACKNOWLEDGMENTS

We thank Sudarshan A. Shetty for his clarifications and input in reproducing the workflow from their previous study (33). We thank Ákos T. Kovács for an insightful discussion on the microbe-microbe interactions occurring in rhizosphere communities.

This work was supported by funding from NWO-XL (OCENW.XL21.XL21.088 to J.J.J.V.D.H and M.H.M.). Contributor roles: Conceptualization, R.A.K., Z.L.R., C.B., J.J.J.V.D.H., and M.H.M.; Methodology, R.A.K., Z.L.R., J.J.J.V.D.H., and M.H.M.; Software, R.A.K. and Z.L.R.; Funding acquisition and supervision, J.J.J.V.D.H. and M.H.M.; Data curation and resources, M.G.C. and J.H.; Formal analysis, investigation, visualization, and writing—original draft, R.A.K.; Writing—review & editing, R.A.K., Z.L.R., C.B., M.G.C., Y.Z., J.J.J.V.D.H., and M.H.M..

During the preparation of this manuscript, ChatGPT was used to improve grammar and readability. All suggestions provided by the AI models were carefully reviewed to ensure that the content remained unchanged.

## AUTHOR AFFILIATIONS

[1]Bioinformatics Group, Wageningen University, Wageningen, the Netherlands

[2]Laboratory of Microbiology, Wageningen University, Wageningen, the Netherlands

[3]Department of Plant Pathology and Wisconsin Institute for Discovery, University of Wisconsin-Madison, Madison, Wisconsin, USA

[4]Institute of Precision Medicine, the First Affiliated Hospital, Sun Yat-sen University, Guangzhou, Guangdong, China

[5]Department of Biochemistry, University of Johannesburg, Johannesburg, South Africa

**PRESENT ADDRESS**

Zachary L. Reitz, Department of Ecology, Evolution and Marine Biology, University of California, Santa Barbara, California, USA

**AUTHOR ORCIDs**

Robert A. Koetsier ⬤ http://orcid.org/0000-0002-4477-5401
Zachary L. Reitz ⬤ http://orcid.org/0000-0003-1964-8221
Clara Belzer ⬤ http://orcid.org/0000-0001-6922-836X
Marc G. Chevrette ⬤ http://orcid.org/0000-0002-7209-0717
Jo Handelsman ⬤ http://orcid.org/0000-0003-3488-5030
Yijun Zhu ⬤ http://orcid.org/0000-0003-2496-5395
Justin J. J. van der Hooft ⬤ http://orcid.org/0000-0002-9340-5511
Marnix H. Medema ⬤ http://orcid.org/0000-0002-2191-2821

**FUNDING**

| Funder | Grant(s) | Author(s) |
|---|---|---|
| Nederlandse Organisatie voor Wetenschappelijk Onderzoek | OCENW.XL21.XL21.088 | Robert A. Koetsier |
| | | Justin J. J. van der Hooft |
| | | Marnix H. Medema |

**AUTHOR CONTRIBUTIONS**

Robert A. Koetsier, Conceptualization, Data curation, Formal analysis, Investigation, Methodology, Resources, Software, Validation, Visualization, Writing – original draft, Writing – review and editing | Zachary L. Reitz, Conceptualization, Data curation, Investigation, Methodology, Software, Writing – review and editing | Clara Belzer, Resources, Validation, Writing – review and editing | Marc G. Chevrette, Data curation, Methodology, Writing – review and editing | Jo Handelsman, Funding acquisition, Resources, Writing – review and editing | Yijun Zhu, Validation, Writing – review and editing | Justin J. J. van der Hooft, Conceptualization, Funding acquisition, Project administration, Supervision, Writing – review and editing | Marnix H. Medema, Conceptualization, Funding acquisition, Project administration, Supervision, Writing – review and editing

**DATA AVAILABILITY**

The genome sequences of *Flavobacterium johnsoniae* strain UW101 (GCA_000016645.1), *Bacillus cereus* strain UW85 (Ga0417192), and *Pseudomonas koreensis* strain CI12 (Ga0417193) are available on NCBI and JGI-IMG/M (81–83). Genome annotations for *Bacillus cereus* UW85, including the locus tags used in Table 2, are available under NCBI RefSeq accession number GCF_024168525.1. RNA-seq data for the MDb-MM can be found in the syncomR GitHub repository (84). RNA-seq reads for the THOR community are available under the NCBI BioProject accession numbers PRJNA885088 and PRJNA1321520 (85, 86). The genome annotations used for the THOR co-expression analysis, as well as the code used for the analyses presented in this manuscript, are available in the GitHub repository accompanying this publication: https://github.com/robert-koetsier/cross_species_coexpression/.

**ADDITIONAL FILES**

The following material is available online.

## Supplemental Material

**Supplemental Figures (mSystems01321-25-S0001.pdf).** Figures S1 to S12.
**Supplemental Tables (mSystems01321-25-S0002.xlsx).** Tables S1 to S6.
**Supplemental File 1 (mSystems01321-25-S0003.pdf).** An overview of the genes that make up the final, refined BGCs investigated in the co-expression analyses. The heatmaps show the co-expression (Pearson correlation) between all the genes of the BGC, 'x' is shown in a cell if a gene was removed from the analysis due to low expression. Letters on the diagonal correspond to the roles that antiSMASH has assigned to the genes of this BGC, B = core biosynthetic gene, A = Additional biosynthetic gene, T = transport-related gene, R = regulatory gene. Black squares around the cells indicate the genes that compose the refined BGC. In the title of the plots, abbreviations show the BGC's organism, Bc = Bacillus cereus, Fj = Flavobacterium johnsoniae, Pk = Pseudomonas koreensis. If two columns of heatmaps are shown, the black square on the left heatmap will indicate what genes were finally used for the BGC, while the black square on the right heatmap indicates which genes are known to be involved with the synthesis of the metabolite based on literature.

## Open Peer Review

**PEER REVIEW HISTORY (review-history.pdf).** An accounting of the reviewer comments and feedback.

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
