## [Reviewer comments · mSystems]

Using cross-species co-expression to predict metabolic interactions in microbiomes.

Robert Koetsier, Zachary Reitz, Clara Belzer, Marc Chevrette, Jo Handelsman, Yijun Zhu, Justin van der Hooft, and Marnix Medema

Corresponding Author(s): Marnix Medema, Wageningen University

Review Timeline:

Submission Date:	September 10, 2025
Editorial Decision:	October 19, 2025
Revision Received:	November 11, 2025
Accepted:	November 13, 2025

Editor: Pablo Ivan Nikel

Reviewer(s): The reviewers have opted to remain anonymous.

Transaction Report:

DOI: <https://doi.org/10.1128/msystems.01321-25>

Re: mSystems01321-25 (Using cross-species co-expression to predict metabolic interactions in microbiomes.)

Dear Prof. Medema:

Dear Marnix:

Thank you for submitting your article to mSystems. Two reviewers had accepted my invitation to comment on your submission, but one of them indicated he/she would not be available after accepting the invitation. To avoid further delays, I am basing my decision on the review that is available and my own evaluation of the article. I am then inviting you to submit a revised version of your paper that addresses the reviewer's comments, which you can find at the end of this letter. Please return the manuscript within 60 days; if you cannot complete the modification within this time period, please contact me. If you do not wish to modify the manuscript and prefer to submit it to another journal, notify me immediately so that the manuscript may be formally withdrawn from consideration by mSystems.

Revision Guidelines

Sincerely, Pablo

Prof. Pablo Ivan Nikel
Editor
mSystems

Reviewer #1 (Comments for the Author):

The manuscript by Koetsier et al describes the use of microbial co-culture RNA sequencing data to assess metabolic interactions. They specifically addressed if co-expression between genes can predict competition, cross-feeding, and specialized metabolite interactions. They do this using existing datasets and determined that cross-species co-expression could be used to predict pathways potentially involved in microbial interactions - as well as capturing known patterns. This seems like a useful new

computational approach to assess metabolic interactions and understand subtle differences among closely related species.

The premise behind the approach is that co-expressed genes are related to the same biological process. One concern about this paper, which the authors mention, is how to distinguish between spurious associations and those that represent real interactions. I think the authors need to make clear early on and throughout that this approach can only generate hypotheses about which co-expression patterns may be meaningful.

Line 191: it is stated that not all interspecies edges could be interpreted in the context of metabolic interactions. Is there a way to infer which ones do not - or to prioritize which ones are most likely to be meaningful other than correlation scores? Here it would be good to say something about next steps - possibly experiments to determine which are relevant?

What if some gene products repress the expression of other genes? Are negative associations also detected? Line 358 is the first mention of negative co-expression.

Minor points: sentence starting line 97 should be revised for clarity.

Figure 1A and B: the first steps are to access existing data. The figure implies that the authors did the co-culturing and RNA isolation.

Line 250: "or whether this finding would generalize" should be revised.

Figure 3: how do we know which histograms correspond to which species (e.g. the Bacteriodes)?

Line 481: ".....to mitigate the effects of confounding." Needs to be revised.

Response to Reviewer remarks

The manuscript by Koetsier et al describes the use of microbial co-culture RNA sequencing data to assess metabolic interactions. They specifically addressed if co-expression between genes can predict competition, cross-feeding, and specialized metabolite interactions. They do this using existing datasets and determined that cross-species co-expression could be used to predict pathways potentially involved in microbial interactions - as well as capturing known patterns. This seems like a useful new computational approach to assess metabolic interactions and understand subtle differences among closely related species.

We thank the reviewer for their positive words and thoughtful comments on the manuscript. Below, we have included point-by-point responses to each of the reviewer's remarks. We hope that these accurately address any questions or concerns raised. Based on the reviewer's suggestions, we have made several edits to the manuscript that are outlined in each of our responses. The suggestions have helped us to more clearly highlight the opportunities and pitfalls of the cross-species co-expression method, as well as to present a more complete discussion of the approach.

The premise behind the approach is that co-expressed genes are related to the same biological process. One concern about this paper, which the authors mention, is how to distinguish between spurious associations and those that represent real interactions. I think the authors need to make clear early on and throughout that this approach can only generate hypotheses about which co-expression patterns may be meaningful.

The relationships identified by our approach indeed do not necessarily imply causality and are sensitive to confounders. To more clearly highlight this caveat, we have added several related remarks along with the paper's interpretations and conclusions. The following revisions were made in the manuscript:

- We emphasized the pitfall in the last paragraph of the introduction. The modified text now reads as follows, beginning at Line 103: "Several of these are likely to be spurious, for example caused by pairs of microbial strains simultaneously responding to environmental changes rather than direct interactions. This is a potential pitfall of the method that should be addressed through careful experimental design, such as including (sufficient numbers of) conditions that specifically elicit responses in individual microbes. Nevertheless, other associations we assess likely represent real interactions."
- We highlighted the issue of spurious associations in the first Results analysis section, line 202: "Spurious associations caused by confounders are a common issue in correlation analyses (39), which can complicate interpretation in untargeted approaches."
- We highlighted importance of experimental validation, line 320: "Again, this finding highlights the utility of cross-species co-expression analysis in identifying microbe pairs with the most compatible production and consumption dynamics. However, this analysis should be complemented by experimental validation, since correlation-based methods alone do not confirm causal relationships."
- We now explicitly mention the spurious associations pitfall and hypothesis generation as an end-goal in the Conclusions section. Line 469: "Our untargeted cross-species co-expression analysis revealed some spurious associations, while the results of the targeted approach effectively recapitulated previously described insights into microbial nutrient utilization dynamics. Furthermore, the method showed a potential

to highlight pairs of microbes with compatible expression dynamics for cross-feeding. Finally, we confirmed that antibiotic response genes are co-expressed with the koreenceine BGC, consistent with the known biological role of this metabolite. These findings confirm that cross-species co-expression can be used to prioritize hypotheses of microbe-microbe interactions, which can be experimentally validated.”

With these edits, we have further emphasized some of the key challenges and the method’s focus on hypothesis generation. These revisions expand upon the originally discussed challenges (e.g., expression \neq metabolite abundance) and remarks, highlighting the limitations of correlation-based approaches.

Line 191: it is stated that not all interspecies edges could be interpreted in the context of metabolic interactions. Is there a way to infer which ones do not - or to prioritize which ones are most likely to be meaningful other than correlation scores? Here it would be good to say something about next steps - possibly experiments to determine which are relevant?

We consider the issue here to be a mixed origin of high correlation scores; some we assume represent “true” microbe-microbe interactions, others represent spurious associations likely caused by environmental confounders. A possible approach to shortlist pathway associations more likely to represent true microbe-microbe interactions involves incorporating prior knowledge to filter for theoretically feasible interactions between gut microbes. Previous research highlights the importance of cross-feeding and competition for nutrient resources for understanding the interactions between gut microbes (1–5). Therefore, pathway associations that could be classified as either cross-feeding or competition interactions could be filtered for to restrict the top interspecies edges to the most plausible subset. In practice, this approach could look very similar to the analyses we conducted to investigate carbohydrate degradation dynamics or cross-feeding based on relations between metabolite-producing and consuming pathways.

The previous version of the manuscript did not explicitly include the recommendation to conduct further filtering or a follow-up analysis specifically targeting pathways that are likely involved in microbe-microbe interactions. We have now revised the section to include this suggestion. This strategy may help reduce some spurious associations, but it would be preferable to design experiments with cross-species co-expression analysis in mind. This would involve aiming to reduce confounders, for example, following some of the suggestions that we have outlined in the conclusions and future perspectives section. To include the above in the manuscript, we have added a paragraph, line 202:

“Spurious associations caused by confounders are a common issue in correlation analyses (39), which can complicate interpretation in untargeted approaches. While the confounders are best minimized through careful experimental design, additional filtering of the interspecies edges based on likely pathway connections may help to prioritize the most probable microbial interactions. For example, one could focus exclusively on associations between pathways involved in potential cross-feeding or competition, as we illustrate in the following sections.”

What if some gene products repress the expression of other genes? Are negative associations also detected? Line 358 is the first mention of negative co-expression.

Negative associations in co-expression analysis can be very informative, which is often overlooked. In the analysis of the first dataset (MDb-MM), we feature many positive associations, which may inadvertently give the impression that negative ones are not relevant. For a more complete discussion, we have included an additional example of a negative association in Line 192: “... *Subdoligranulum variabile* valine or isoleucine

biosynthesis vs *Ruminococcus bromii* pyruvate:ferredoxin oxidoreductase ($r = -0.90$). Apart from this, the original version of the manuscript already featured several examples of negative associations earlier in the text. E.g., line 182: “dissimilar expression of the species’ GMMs for allose degradation ($r = -0.31$) and arabinoxylan degradation ($r = -0.15$) indicates a possible nutritional complementarity that facilitates co-existence (Table S2).”; Table 1; and line 258 (analysis into general trends): “No significant negative shifts were detected.”. The discussion of negative associations in the section on cross-feeding is minimal, since their interpretation in this context is not so intuitive. A generalized example for illustration of this unintuitive aspect: microbe A expresses genes to utilize metabolite X, while microbe B does not express the genes to produce metabolite X. As the reviewer correctly highlighted, the analyses into THOR community interactions contain more extensive evaluation of negative correlations.

Minor points: sentence starting line 97 should be revised for clarity.

The sentence has been simplified for clarity. Revised version, line 94: “It would be desirable to further assess the predictions of cross-species co-expression using multiple datasets that include strains with known reference information.”

Figure 1A and B: the first steps are to access existing data. The figure implies that the authors did the co-culturing and RNA isolation.

We have updated Figure 1 to clearly indicate that public datasets served as the starting point for our analyses. The revised figure, shown below, now includes database and download icons, as well as “Public RNA-seq dataset download” descriptions where appropriate.

Line 250: "or whether this finding would generalize" should be revised.

Now line 250, sentence revised to:

“By calculating the co-expression between all pairs of equivalent GMMs from different microbes, we aimed to assess whether low to moderate similarity in expression profiles was unique to fructose and starch degradation or generally observed among carbohydrate degradation GMMs.”

Figure 3: how do we know which histograms correspond to which species (e.g. the Bacteriodes)?

We acknowledge that this link can indeed be challenging to make from the histograms alone. The information is available in Table S2, which we had not yet referenced in the caption of Figure 3. We have included this reference in the revised version of the manuscript for better traceability of the histogram’s underlying data.

Line 481: ".....to mitigate the effects of confounding." Needs to be revised.

Revised, line 487:

“Careful selection of conditions included in the experimental design can help to reduce confounders.”

Bibliography

1. Culp EJ, Goodman AL. 2023. Cross-feeding in the gut microbiome: Ecology and mechanisms. *Cell Host Microbe* 31:485–499.
2. Gutiérrez N, Garrido D. 2019. Species Deletions from Microbiome Consortia Reveal Key Metabolic Interactions between Gut Microbes. *mSystems* 4:10.1128/msystems.00185-19.
3. Pereira FC, Berry D. 2017. Microbial nutrient niches in the gut. *Environ Microbiol* 19:1366–1378.
4. Shetty SA, Kuipers B, Atashgahi S, Aalvink S, Smidt H, de Vos WM. 2022. Inter-species Metabolic Interactions in an In-vitro Minimal Human Gut Microbiome of Core Bacteria. *NPJ Biofilms Microbiomes* 8:21.
5. Wang T, Goyal A, Dubinkina V, Maslov S. 2019. Evidence for a multi-level trophic organization of the human gut microbiome. *PLOS Comput Biol* 15:e1007524.

Re: mSystems01321-25R1 (Using cross-species co-expression to predict metabolic interactions in microbiomes.)

Dear Prof. Medema, dear Prof. van der Hoof:
Dear Marnix, dear Justin,

Thanks for submitting your revised manuscript to mSystems. I am delighted to report that I am accepting your work for publication in our journal, and I am forwarding it to the ASM production staff for publication. Your paper will first be checked to make sure all elements meet the technical requirements. ASM staff will contact you if anything needs to be revised before copyediting and production can begin. Otherwise, you will be notified when your proofs are ready to be viewed.

Sincerely, Pablo

Prof. Pablo Ivan Nikel
Editor
mSystems